# Associations between continuity of primary and specialty physician care and use of hospital-based care among community-dwelling older adults with complex care needs

**Aaron Jones**[1]*, **Susan E. Bronskill**[2,3], **Hsien Seow**[1,4], **Mats Junek**[1], **David Feeny**[5], **Andrew P. Costa**[1,6]

1 Department of Health Research Methods, Evidence, and Impact, McMaster University, Hamilton, Ontario, Canada, 2 ICES, Toronto, Ontario, Canada, 3 Institute of Health Policy, Management & Evaluation, Dalla Lana School of Public Health, University of Toronto, Ontario, Canada, 4 Department of Oncology, McMaster University, Hamilton, Ontario, Canada, 5 Department of Economics, McMaster University, Hamilton, Ontario, Canada, 6 Department of Medicine, McMaster University, Hamilton, Ontario, Canada

* jonesa13@mcmaster.ca

**Data Availability Statement:** The dataset used in this study is held securely in coded format at ICES. ICES is a prescribed entity under section 45 of

## Abstract

### Objective

While research suggests that higher continuity of primary and specialty physician care can improve patient outcomes, their effects have rarely been examined and compared concurrently. We investigated associations between continuity of primary and specialty physician care and emergency department visits and hospital admissions among community-dwelling older adults with complex care needs.

### Methods

We conducted a retrospective cohort study of home care patients in Ontario, Canada, from October 2014 to September 2016. We measured continuity of primary and specialty physician care over the two years prior to a home care assessment and categorized them into low, medium, and high groups using terciles of the distribution. We used Cox regression models to concurrently test the associations between continuity of primary and specialty care and risk of an emergency department visit and hospital admission within six months of assessment, controlling for potential confounders. We examined interactions between continuity of care and count of chronic conditions, count of physician specialties seen, functional impairment, and cognitive impairment.

### Results

Of 178,686 participants, 49% had an emergency department visit during follow-up and 27% had a hospital admission. High vs. low continuity of primary care was associated with a reduced risk of an emergency department visit (HR = 0.90 (0.89–0.92)) as was continuity of specialty care (HR = 0.93 (0.91–0.95)). High vs. low continuity of primary care was associated also with a reduced risk of a hospital admission (HR = 0.94 (0.92–0.96)) as was

Ontario's Personal Health Information Protection Act. Section 45 authorizes ICES to collect personal health information, without consent, for the purpose of analysis or compiling statistical information with respect to the management of, evaluation or monitoring of, the allocation of resources to or planning for all or part of the health system. Legal restrictions and data sharing agreements prohibit ICES from making the dataset publicly available to protect potentially identifiable personal health information. However, access to the dataset may be granted to those who meet pre-specified criteria for confidential access, available at https://www.ices.on.ca/das. The full dataset creation plan and underlying analytic code have been included as supplemental information files.

**Funding:** AC received a grant for this research from the Canadian Institutes of Health Research (148933) (https://cihr-irsc.gc.ca/e/193.html). This study was supported by ICES (www.ices.on.ca), which is funded by an annual grant from the Ontario Ministry of Health and Long-Term Care (MOHLTC). The analyses, conclusions, opinions and statements expressed herein are solely those of the authors and do not reflect those of the funding or data sources; no endorsement is intended or should be inferred. Parts of this material are based on data and/or information compiled and provided by the Canadian Institute for Health Information (CIHI)(www.cihi.ca). However, the analyses, conclusions, opinions and statements expressed in the material are those of the authors, and not necessarily those of CIHI. The funders had no role in study design, data collection and analysis, decision to publish, or preparation of the manuscript.

**Competing interests:** The authors have declared that no competing interests exist.

continuity of specialty care (HR = 0.92 (0.90–0.94)). The effect of continuity of specialty care was moderately stronger among patients who saw four or more physician specialties.

## Conclusion

Higher continuity of primary physician and specialty physician care had independent, protective effects of similar magnitude against emergency department use and hospital admissions. Improving continuity of specialty care should be a priority alongside improving continuity of primary care in complex, older adult populations with significant specialist use.

## Introduction

Global population aging has resulted in a growing number of older adults living in the community with complex care needs such as multimorbidity, functional impairment, and frailty [1,2]. Global estimates of multimorbidity among older adults exceeds 50% [3], with estimates as high 81% in the United States [4], and figures are expected to continue to rise in the future [5–7]. The intensity of emergency department visits, hospitalizations, and overall health care expenditure increases with older age, and are further exacerbated by factors such as multimorbidity and frailty [4,8–10]. The growing challenge of multimorbidity and other complex care needs among older adults have spurred calls for a larger interdisciplinary physician workforce of both primary care and specialty care physicians, and greater continuity of physician care [7,11,12]

Continuity of care has been studied within health services research for decades as a method of examining how patients interact with their health care providers. Continuity is a complex construct with multiple aspects, including information continuity, management continuity, and interpersonal (or relational) continuity, the last of which is concerned with characterizing the on-going relationship between patient and provider [13]. A necessary component of interpersonal continuity is longitudinal continuity, which refers to the consistency with which a patient visits the same health care providers over time [14]. A continuous, longitudinal relationship between a provider and patient has been shown to foster trust and familiarity, which can yield multiple benefits such as increased adherence to care plans, more effective communication, and greater satisfaction in care [15,16]. Higher continuity of care with physicians has been consistently linked to positive outcomes such as fewer emergency department visits, fewer hospital admissions, and lower mortality [17–19]. Consequently, improving continuity of care is a frequently sought objective of health care systems [20–22].

The development of the patient-physician relationship through longitudinal continuity has traditionally been highly valued within primary care [13,23]. More recently, the measurement and assessment of continuity within other physician specialties has become a topic of interest, although research is still limited [24–26]. Additionally, some researchers have examined continuity across all specialties (including primary care), particularly for multimorbid or otherwise complex patients who are expected to receive a significant portion of their care from specialist physicians [27–30]. In general, research suggests that continuity of both primary care and specialty physician care improve health utilization and mortality outcomes [17,31]. However, there has been little research that has concurrently examined and compared the effects of continuity of primary and specialty physician care in populations that are significant users of both types of care. Knowledge of the relative effectiveness of continuity of primary and specialty

care can help inform strategies to promote continuity of care for older adults with complex care needs.

The objective of this study is to examine and compare the associations between continuity of primary and specialty physician care and emergency department use and hospital admissions and to explore potential modification of the effects of continuity. Within a cohort of community-dwelling older adults with complex care needs, we will determine whether continuity of primary and specialty care have independent effects, the relative magnitude of those effects, and examine interactions between continuity of care and increasing multimorbidity, use of physician specialties, functional impairment, and cognitive impairment.

## Methods

### Setting

Ontario is Canada's most populous province, with an estimated population of 13.7 million in 2015, including 3 million residents aged 60 years or older. Most residents are covered by Ontario's universal, publicly-funded, health insurance program that covers medically necessary services, including physician care, hospital and emergency department care, home care, and other services. Ontario operates a "gatekeeper" system in which access to specialist physicians requires a referral from primary care physician. Ontario offers publicly-funded home care for eligible residents which may include nursing, personal support and homemaking, physiotherapy, occupational therapy, and other services. Eligibility is based on need and criteria typically include difficulty in performing activities of daily living (such as bathing or toileting) or need for frequent nursing for reasons such as wound care, catheter/ostomy care, intravenous medications, or chronic disease monitoring.

**Study design, population, and data sources.** We conducted a population-based, retrospective cohort study of older adults receiving home care on an on-going basis in Ontario, Canada. Home care patients in Ontario are typically community-dwelling older adults characterized by multiple chronic conditions and/or functional and cognitive impairments. We focused on home care patients as the availability of accurate clinical measures, significant use of primary and specialist physicians, and frequent emergency department visitation make them an ideal population in which to examine the simultaneous influence of continuity of primary and specialty physician care [32]. We used multiple, linked, health administrative databases to identify a cohort of older adult home care patients who received a comprehensive home care assessment. Home care patients were identified using the Home Care Database. Physician billing claims were extracted from the Ontario Health Insurance Plan database. The National Ambulatory Reporting System was used to identify emergency department visits and the Discharge Abstract Database was used to capture hospital admissions Patient deaths were identified with the Registered Persons Database and admission to long-term care homes with the Continuing Care Reporting System. Datasets were linked using unique encoded identifiers and analyzed at ICES (S1 Appendix). This study was granted an exemption from formal ethics review by the Hamilton Integrated Research Ethics Board as the use of data in this project was authorized under section 45 of Ontario's Personal Health Information Protection Act, which does not require review by a research ethics board.

**Participants.** Home care patients receiving on-going home care in Ontario are frequently assessed with the Resident Assessment Instrument for Home Care (RAI-HC) [33], which is a comprehensive clinical assessment. The reliability and validity of the RAI-HC assessment is well documented. [34–36] We selected all RAI-HC assessments for publicly-funded home care patients aged 60 years or older that were completed in Ontario between October 1, 2014 and September 30, 2016. If an individual was assessed more than once during the accrual period,

their most recent assessment was used. This assessment date was used as the reference date for cohort entry. To ensure that both continuity of primary and specialty care could be calculated for all participants, we included only patients with at least two primary care physician visits and two specialist physician visits (within the same specialty) in the two years prior to the assessment.

## Measures

**Modified Bice-Boxerman continuity of care index.** The Bice-Boxerman continuity of care index measures the dispersion of health care visits among providers, reaching a maximum value of one when all visits are within one provider and a minimum value of zero when all visits are to different providers [37]. The index is one of the most commonly used measures of longitudinal continuity and has been employed within single physician specialties as well as across multiple specialties [38]. However, using the Bice-Boxerman index across multiple physician specialties results in lower continuity for patients who see more than one specialty as the physicians operating within the different specialties will naturally be different. The more physicians from different specialties a patient sees, the lower their continuity will be. Moreover, patients with complex care needs may benefit from regularly seeing physicians from multiple specialties, meaning that higher continuity when measured in the traditional manner may neither be desirable or optimal for these patients [39]. This complicates the interpretation of the Bice-Boxerman index, as higher continuity may no longer be expected to be associated with improved patient outcomes.

To address these limitations and preserve the expectation that higher continuity should be associated with improved outcomes, we modified the Bice-Boxerman index to focus on fragmentation of care within each specialty rather than across specialties. Our modified version divides the original Bice-Boxerman index by the maximum value of the index each patient could achieve assuming that each visit within each specialty was to the same physician. The resulting modified index reaches a maximum value of one when all visits *within each specialty* are to the same physician and a value of zero when each visit is to a different physician. The modified index is identical to the original index when only one specialty considered and is otherwise equivalent to a weighted average of specialty-specific Bice-Boxerman indices, assuming that specialty included has least two visits. The formulae for the original and modified Bice-Boxerman indices, along with an empirical example and proof can be found in S2 Appendix.

We used the modified Bice-Boxerman index to calculate continuity of care separately for primary care and specialty care. For primary care physician continuity, we included all ambulatory physician visits in the two years prior to the baseline assessment within family practice/general practice and community medicine (Fig 1). For specialty physician continuity we included all ambulatory visits in the two years prior to the baseline assessment from all remaining physician specialties. For use in statistical analysis we split the continuity indices into high, medium, and low groups based on terciles of the sample distribution.

**Outcomes.** Associations between continuity of care and use of hospital-based care are among the frequently tested hypotheses in the literature on continuity of care [18]. Home care patients have been previously noted to have high rates of emergency department visits and hospital admissions, which contribute to health system overcrowding may lead adverse events such as delirium and deconditioning [40,41]. We followed patients for six months after the baseline assessment and calculated the number of days until the first emergency department visit and number of days until the first hospital admission as our primary outcomes. The outcomes were censored at date of death, admission to a long-term care home, and at the end of the six-month follow-up window.

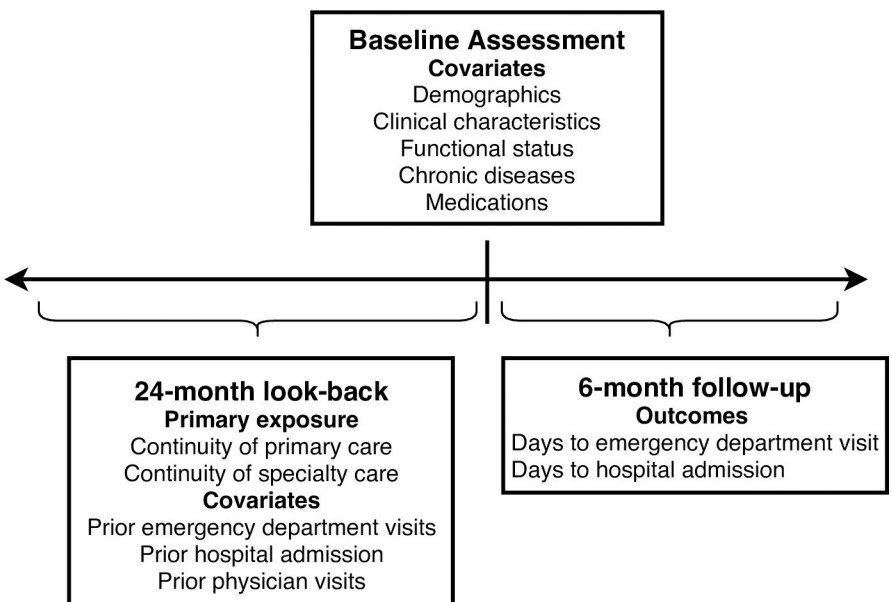

**Fig 1. Study timeline.**

Covariates.  We identified important covariates to adjust for confounding in statistical models based on previous research [27,42]. These covariates included age, sex, rurality, count of chronic conditions, count of physician specialties seen in the previous two years (including primary care), congestive heart failure, chronic obstructive pulmonary disease, count of concurrent medications, count of outpatient physician visits in previous two years, count of emergency department visits in the previous two years, and hospital admission in the previous two years. Chronic diseases and medications were measured using the baseline RAI-HC assessment. All other covariates were extracted from administrative data sources. We focused on congestive heart failure and chronic obstructive pulmonary disease in particular as they have been shown to be major risk factors for use of hospital-based care in home care patients [43]. Our broader count of chronic conditions included: stroke, congestive heart failure, hypertension, dementia, Parkinsonism, multiple sclerosis, arthritis, osteoporosis, any psychiatric condition, cancer, chronic obstructive pulmonary disease, diabetes, and renal failure. The count of physician specialties only included those specialties in which a patient had a least two visits in the past two years to align with our calculation of continuity of care.

Count of chronic conditions, count of physician specialties, functional impairment, and cognitive impairment were identified as potential modifiers of the relationship between continuity of care and emergency department use. To examine modification across count of chronic conditions and count of physician specialties, we categorized each variable into three groups, as equally-sized as possible, based on the sample distribution. Functional impairment was measured using the ADL Hierarchy Scale (ADL) [44] and split into 3 categories, 0–1, 2–3, 4–6. Cognitive impairment was measured by the Cognitive Performance Scale [45](CPS) and also split into 3 categories: 0–1, 2–3, 4–6.

## Analysis

We reported the demographic and health characteristics our of cohort. We further described the distribution of each continuity index, physician use within the two years prior to the

baseline assessment, and proportion patients with an emergency department visit and hospital admission during follow-up. We used multivariable Cox regression models to examine the associations between continuity of primary care and continuity of specialty care and risk of each outcome, controlling for identified confounders. To examine effect modification, we fit additional models with interaction terms between the continuity of care measures each of our potential effect modifiers. We reported the hazard ratios and 95% confidence intervals of an emergency department visit and hospital admission for all variables in the initial Cox models. For the effect modification models, we reported the hazard ratios and 95% confidence intervals for high vs. low continuity of primary and specialty care within each category of the effect modifiers and the p-value of the interaction term.

## Results

Of the 232,694 unique older adults home care patients with a RAI-HC assessment, 178,686, patients had at least two primary care physician visits and at least two specialist physician visits (within the same specialty) during the two years prior to the assessment. The median age of the population was 82 years and 61% were female (Table 1). Over half (59%) of the patients had at least a mild cognitive impairment (CPS $> = 2$) and 42% needed at least limited assistance with the activities of daily living (ADL $> = 2$). The most common chronic conditions were hypertension (66%), arthritis (54%) and diabetes (30%). The median number of chronic conditions was three. The proportion of patients with an emergency department visit during the six-month follow-up was 49% while 27% had a hospital admission.

### Distribution of continuity indices and baseline physician use

The median value of continuity of primary care was 0.73 (Table 2). The 33th and 66th percentiles used to define the low, medium, and high continuity of primary care groups were 0.54, and 0.88 respectively. The median value of the continuity of specialty care was 0.89 and the 33th and 66th percentiles used to define the low, medium, and high continuity of specialty care groups were 0.68, and 1. The median count of physician visits in the two years prior to the baseline assessment was 27, with a median of 14 visits within primary care and 10 visits within specialty care.

### Association between continuity of care and emergency department visits

Both continuity of primary and specialty physician care were associated with small reductions of generally similar size in the risk of an emergency department visit (Table 3). High vs. low continuity of primary care was associated with an a hazard ratio (HR) of 0.90 (95% CI 0.89–0.92) while medium vs. low continuity was associated with an HR of 0.96 (95% CI 0.94–0.98). High vs. low continuity of specialty care was associated with a HR of 0.93 (0.91–0.95) while medium vs. low continuity was associated with HR of 0.97 (0.95–0.99).

### Association between continuity of care and hospital admissions

Continuity of primary and specialty physician care were also both associated with small reductions in the risk of a hospital admission (Table 3). High vs. low continuity of primary care was associated with an HR of 0.94 (95% CI 0.92–0.96) while medium vs. low continuity was associated with an HR of 0.96 (95% CI 0.94–0.98). High vs. low continuity of specialty care was associated with a HR of 0.92 (0.90–0.94) while high vs. medium continuity was associated with an HR of 0.96 (0.94–0.99).

**Table 1. Baseline characteristics of cohort members.**

| | no. (%) |
|---|---|
| **Patient Characteristics** | **n = 178,686** |
| **Demographics** | |
| Age, yr (Median (Q1, Q3)) | 82 (75, 88) |
| Sex, female | 109620 (61) |
| Lived Alone | 80436 (45) |
| Rurality | |
| Urban | 121161 (71) |
| Semiurban | 38584 (22) |
| Rural | 13763 (8) |
| **Health** | |
| ADL Impairment[a] | |
| Independent/Supervision | 104872 (59) |
| Limited/Extensive | 54468 (31) |
| Maximal/ Dependent | 19168 (11) |
| Cognitive Impairment[b] | |
| Intact / Borderline intact | 72910 (41) |
| Mild / Moderate | 93527 (52) |
| Severe | 12071 (7) |
| Number of Medications | |
| 0–4 | 21754 (12) |
| 5–8 | 54722 (31) |
| 9 or more | 102032 (57) |
| Any mood symptom | 92340 (52) |
| Bladder incontinence | 71017 (40) |
| Fall in last 90 days | 75309 (42) |
| **Chronic Conditions** | |
| Congestive heart failure | 27043 (15) |
| Stroke | 31319 (18) |
| Hypertension | 117952 (66) |
| Chronic obstructive pulmonary disease | 36681 (21) |
| Diabetes | 53990 (30) |
| Dementia | 43211 (24) |
| Multiple Sclerosis | 1609 (1) |
| Parkinsonism | 9674 (5) |
| Arthritis | 96309 (54) |
| Osteoporosis | 42713 (24) |
| Psychiatric diagnosis | 34061 (19) |
| Cancer | 31221 (17) |
| Renal failure | 17854 (10) |
| Count of chronic conditions (Median (Q1, Q3)) | 3 (2, 4) |

ADL = Activities of daily living, Q1 = Quartile 1, Q3 = Quartile 3

[a] ADL Hierarchy Scale: Includes personal hygiene, locomotion, eating and toileting

[b] Cognitive performance scale

**Table 2. Distribution of continuity indices and baseline physician utilization.**

| Measure | Median (Q1, Q3) |
|---|---|
| Continuity of primary care | 0.73 (0.47, 1) |
| Continuity of specialty care | 0.89 (0.57,1) |
| Count of physician visits | 27 (17, 40) |
| Count of primary care physician visits | 14 (8, 22) |
| Count of specialty care physician visits | 10 (6, 18) |
| Count of physician specialties seen | 3 (2, 5) |

Covers two years prior to cohort entry

## Effect modification of associations between continuity and emergency department use and hospital admissions

Count of chronic conditions was categorized into groups of 0–2, 3, and 4+ conditions while count physician specialties seen was categorized into 2, 3 and 4+ specialties. Significant modification of the effect of high vs. low continuity of specialty physician care occurred across categories of the number of specialties seen for both outcomes (Figs 2 and 3). The HR of an emergency department visit associated with high vs. low continuity of specialty care was 0.94 (0.91–0.97) for two specialties, 0.96 (0.93–0.99) for three specialties and, 0.90 (0.88–0.93) for four or more specialties. For hospital admissions, the HR associated with high vs. low continuity of specialty care was 0.96 (0.93–1.00) for two specialties, 0.94 (0.90–0.98) for three specialties, and 0.87 (0.84–0.90) for four or more specialties.

Significant modification also occurred in the association between high vs. low continuity of primary care and emergency department visits across categories of cognitive impairment, with the effect of continuity being stronger among patients with a CPS of 0–1 (HR: 0.89 (0.86–0.91)) than those with a CPS of 2–3 (HR: 0.93 (0.91–0.95)) and CPS of 4–6 (HR: 0.93 (0.87–0.99)). However, there was no significant modification for hospital admissions. Finally, there was modification in the association between high vs. low continuity of specialty care and hospital admissions across count of chronic conditions, but this is the result of a substantively weaker association in the middle category of chronic conditions (HR: 0.97 (0.93–1.01) compared to the higher (HR: 0.91 (0.88–0.94)) and lower categories (HR: 0.90 (0.87–0.94)). The lack of a dose-response relationship limits interpretation of this effect.

## Discussion

We found that higher longitudinal continuity of primary physician care and specialty physician care were independently associated with lower risks of emergency department visits and hospital admissions in a population of community-dwelling older adults with complex care needs. The observed risk reductions were small and of generally similar size across continuity measures and outcomes. While there was no consistent modification of the effect of either continuity of primary or specialty care with increasing multimorbidity, the effect of continuity of specialty care was moderately stronger in patients who saw four or more physician specialties. There was also some support for a stronger effect of continuity of primary care among patients without cognitive impairment.

While research suggests that both primary care and specialty physician care are effective at improving patient outcomes, few studies have examined both in the same population in a way that would allow for an assessment of the relative magnitude of their effects. One study by Bayliss et al [31] examined the effects of both primary and specialty physician care in a group of

**Table 3. Hazard ratios and 95% confidence intervals from multivariable Cox models.**

| | Emergency Department Visit | Hospital Admission |
|---|---|---|
| Variable | HR (95%CI) | HR (95%CI) |
| Continuity of primary care | | |
| High | 0.90 (0.89–0.92) | 0.94 (0.92–0.96) |
| Medium | 0.96 (0.94–0.98) | 0.96 (0.94–0.98) |
| Low (ref) | - | - |
| Continuity of specialty care | | |
| High | 0.93 (0.91–0.95) | 0.92 (0.90–0.94) |
| Medium | 0.97 (0.95–0.99) | 0.96 (0.94–0.99) |
| Low (ref) | - | - |
| Sex, F | 0.92 (0.81–0.84) | 0.75 (0.74–0.77) |
| Age | | |
| 60–69 (ref) | - | - |
| 70–79 | 1.01 (0.98–1.03) | 1.04 (1.01–1.07) |
| 80–89 | 1.04 (1.02–1.06) | 1.09 (1.06–1.12) |
| 90+ | 1.18 (1.15–1.20) | 1.30 (1.26–1.34) |
| Rurality | | |
| Urban (ref) | - | - |
| Semiurban | 1.21 (1.19–1.23) | 1.14 (1.11–1.16) |
| Rural | 1.41 (1.38–1.45) | 1.23 (1.20–1.28) |
| Count of comorbid conditions | | |
| 0–2 (ref) | - | - |
| 3 | 1.04 (1.02–1.06) | 1.05 (1.02–1.07) |
| 4+ | 1.12 (1.10–1.14) | 1.13 (1.10–1.16) |
| Count of physician specialties seen | | |
| 2 (ref) | - | - |
| 3 | 1.02 (1.00–1.04) | 1.00 (0.97–1.03) |
| 4+ | 1.09 (1.07–1.12) | 1.07 (1.04–1.10) |
| Congestive heart failure | 1.19 (1.17–1.21) | 1.34 (1.31–1.37) |
| Chronic obstructive pulmonary disease | 1.13 (1.11–1.15) | 1.18 (1.15–1.21) |
| Count of concurrent medications | 1.01 (1.01–1.02) | 1.02 (1.02–1.03) |
| Outpatient physician visits in past two years | 1.00 (1.00–1.01) | 1.00 (1.00–1.00) |
| Emergency department visits in past two years | 1.03 (1.03–1.03) | 1.01 (1.01–1.02) |
| Hospital admission in past two years | 1.45 (1.43–1.47) | 1.75 (1.72–1.78) |

seniors with chronic conditions and concluded that continuity of primary care, but not specialty care, was associated with a reduction in the risk of an emergency department visit. While our finding of similar, independent, effects stands in contrast to the findings of this previous study, our study was conducted in a different population within a different health system and benefited from a considerably larger study size. The previous study also recorded a substantially lower continuity of specialty care than we observed, a difference which is likely related to our use of a modified Bice-Boxerman index that aggregates continuity within each specialty rather than across multiple specialties. Our modified continuity index provides a clearer interpretation when measuring continuity across multiple specialties as it only discounts continuity due to inconsistency in seeing the same physicians within a specialty, rather than being influenced by the overall number of physician specialties seen.

It is reasonable to expect that the associations between continuity of primary and specialty physician care and use of hospital-based care could change with increasing multimorbidity

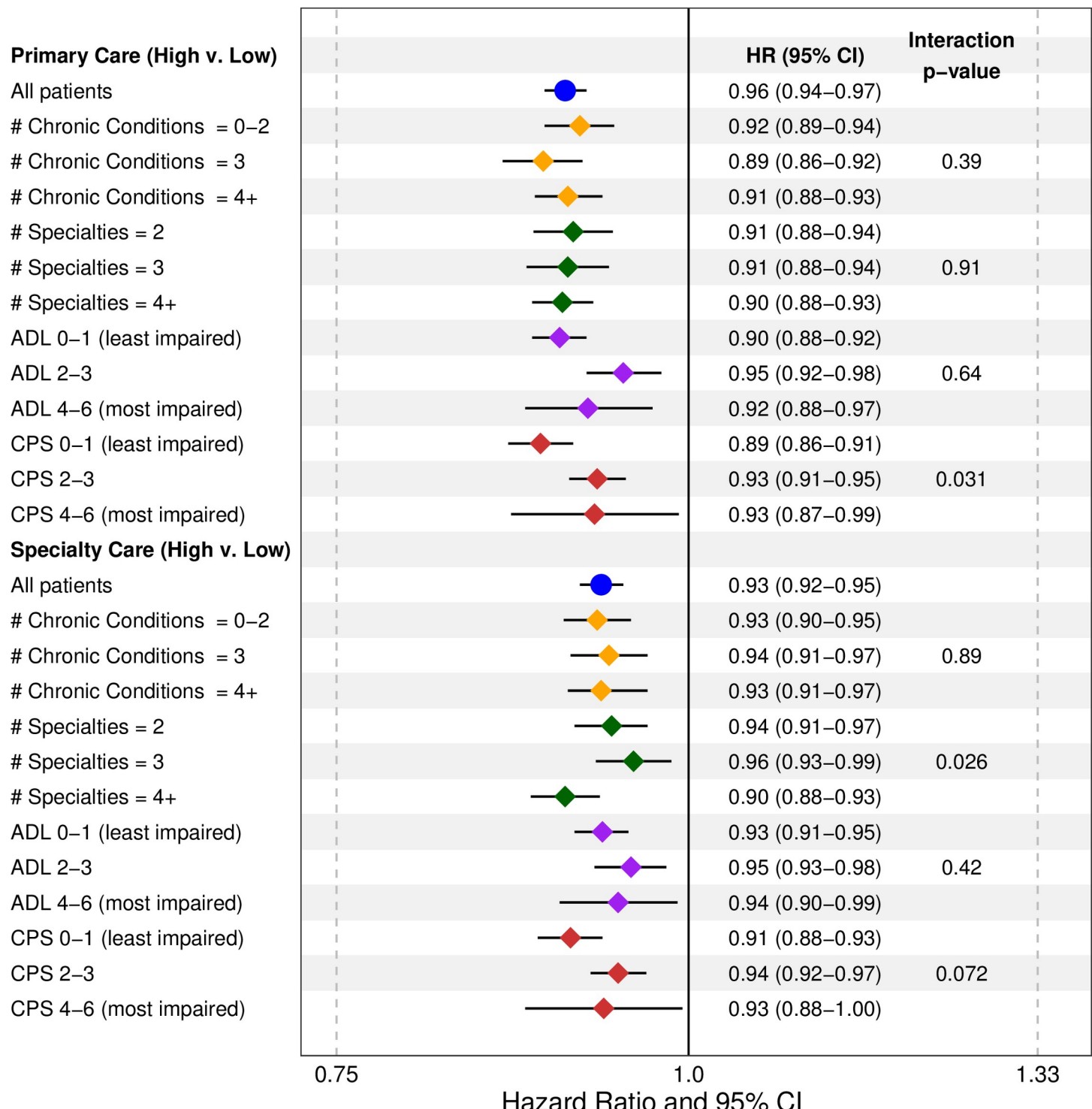

**Fig 2. Associations between continuity of care and risk of an emergency department visit across effect modifiers.**

and use of physician specialties. Multimorbidity presents significant challenges to effectively managing care, and better continuity of care has often been cited as a partial remedy [46,47].

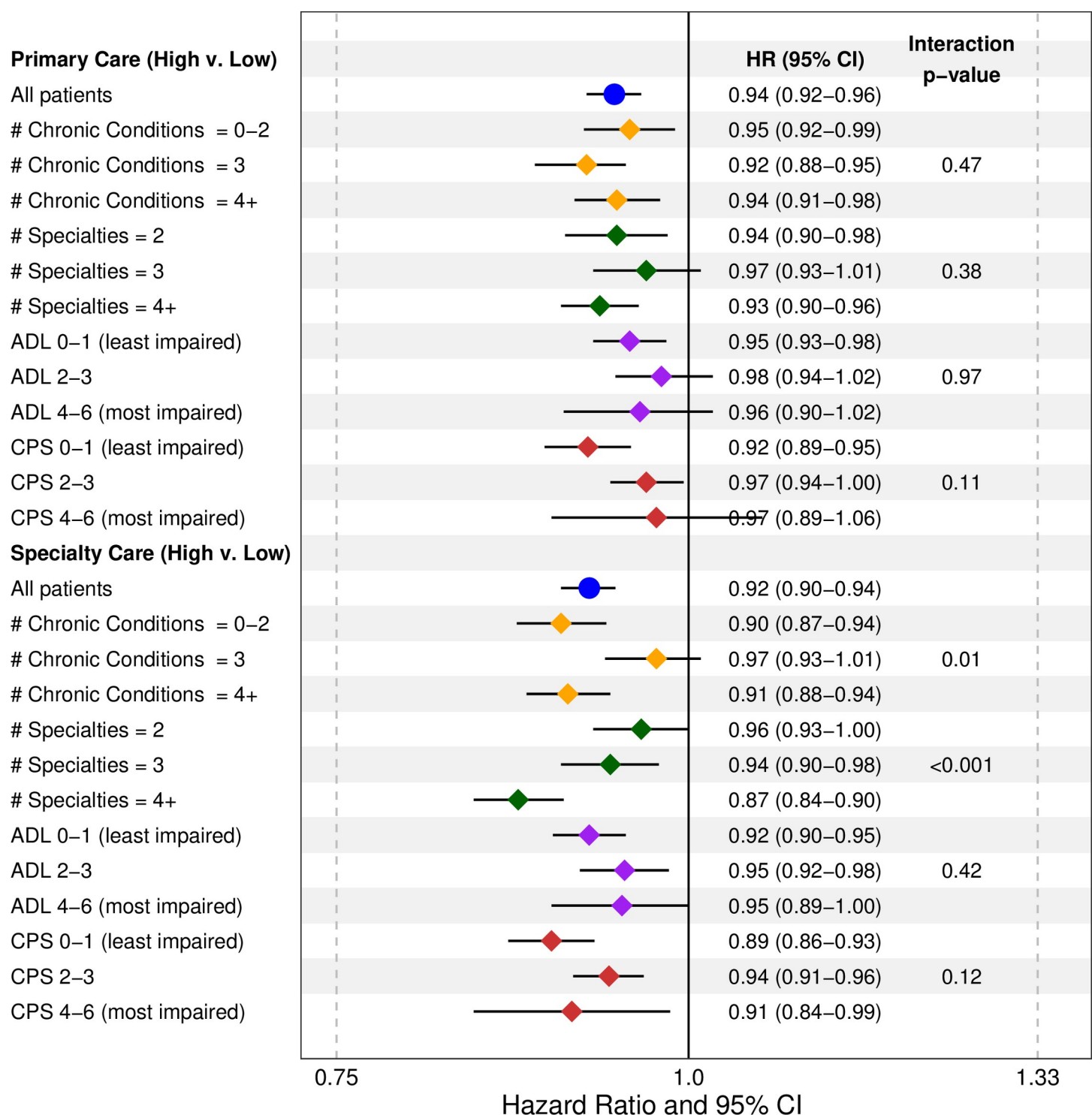

**Fig 3. Associations between continuity of care and risk of a hospital admission across effect modifiers.**

Additionally, it is plausible to imagine that the influence of continuity of specialty care would increase along with the number of physician specialties a patient sees. At the same time,

however, it can be beneficial for patients that see many physicians to have a designated primary care physician at the center that can operate within a patient-centric rather than disease-centric approach and connect with all the other providers [48]. Ultimately, the only significant modification we found was with respect to the effect of continuity of specialty care among patients who saw four or more physician specialties.

While it is intuitive that higher continuity of specialty care is more effective among patients who see more physician specialties, it is intriguing that we found no meaningful modification of the effect of continuity by the count of chronic conditions. Considerable attention has been given to promoting continuity among patients with multimorbidity and research has shown that continuity matters more to patients with more chronic conditions. [49] However, a study by Mondor et al [27] among home care patients with dementia in Ontario found that the association between multimorbidity and emergency department visits did not vary across categories of continuity of care. Another study by Weir et al [30] found that multimorbidity did not meaningfully modify the effect of continuity on hospitalizations and mortality among US patients with incident diabetes. It is also possible that there is a ceiling effect to the influence of multimorbidity on continuity, and that by virtue of being a home care recipient, our population was already in poor enough health to have reached it.

We found no evidence of effect modification of continuity of care across categories of functional impairment, but there was some support for greater effectiveness of continuity of primary care among patients with intact cognition. This modification was only significant in one of our outcomes but the observed hazard ratios trended in the same direction for both measures of continuity in both outcomes. It is intuitive that the relational benefits of increased continuity of care could be lessened for patients with significant cognitive impairments and future research should explore this topic further.

Our findings support the value of consistency in seeing the same specialist physicians alongside consistency in seeing the same primary care physician. While the importance of explicitly considering specialty physicians in informational and management continuity measures has been recognized, much of the attention directed towards improving longitudinal continuity has remained focused on primary care [50,51]. Our results suggest that for complex, older adult populations, efforts to improve the continuity of specialty care should be a priority alongside continuity of primary care. Furthermore, we found that it was not among patients with more chronic conditions, but rather among those who saw more physician specialties, in which continuity of specialty care had a stronger effect [12]. While there is a clear connection between multimorbidity and use of more physician specialties [52], it may be that the additional benefit of continuity of specialty care only incurs when the growing burden of chronic diseases results in visits to a substantial number of physician specialties. Therefore, patients who see numerous physician specialties in additional to their primary care physician should be recognized as key population in which to promote continuity of specialty care.

## Limitations

Our study has several key strengths, including use of population-based data and a large study size. There are, however, notable weaknesses. We used claims-based data to examine longitudinal continuity of care, which is only one aspect of continuity. While the consistency with which a patient sees the same provider is a critical aspect of continuity of care, we were unable to consider other aspects such as informational or management continuity. In complex patients who see multiple physician specialties, the interaction between physicians is clearly of vital importance [50,53]. However, our data sources, similar to other as claims databases, did not contain information on quantity or quality of communication between physicians. Also,

we only examined patients who had at least two primary care and two specialty care physician visits. While this was necessary in order to examine the relative effects of primary and specialty physician care, we cannot generalize some of the other findings, such as the lack of modifying effect by increasing multimorbidity, to a population that does not have any specialist physician use.

## Conclusion

Among community-dwelling older adults with complex care needs, higher longitudinal continuity of primary physician care and specialty physician care had similar, independent, protective effects against emergency department use and hospital admissions. These effects did not vary with increasing multimorbidity, but continuity of specialty physician care was more effective in patients who saw four or more physician specialties. Continuity of specialty physician care should be considered of similar value to continuity primary care among complex, community-dwelling older adults with significant specialist physician use. Patients who see physicians within numerous specialties should be recognized as a group in which continuity of specialty care is of particular importance.

## Supporting information

**S1 Appendix. Databases used in the study.**
(PDF)

**S2 Appendix. Formulae, empirical example, and proof regarding the Bice-Boxerman and modified Bice-Boxerman continuity of care indices.**
(PDF)

**S1 Document. Dataset creation plan.**
(DOCX)

**S2 Document. Analytic code.**
(TXT)

## Author Contributions

**Conceptualization:** Aaron Jones, Mats Junek, Andrew P. Costa.

**Data curation:** Aaron Jones.

**Formal analysis:** Aaron Jones.

**Funding acquisition:** Andrew P. Costa.

**Investigation:** Aaron Jones.

**Methodology:** Aaron Jones, Susan E. Bronskill, Hsien Seow, David Feeny, Andrew P. Costa.

**Supervision:** Susan E. Bronskill, Andrew P. Costa.

**Visualization:** Aaron Jones.

**Writing – original draft:** Aaron Jones.

**Writing – review & editing:** Aaron Jones, Susan E. Bronskill, Hsien Seow, Mats Junek, David Feeny, Andrew P. Costa.

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
