## [Decision Letter · Decision Letter 0]

Transfer alertThis paper was transferred from another journal. As a result, its full editorial history (including decision letters, peer reviews and author responses) may not be present.

23 Apr 2020

PONE-D-20-00490

Associations between continuity of primary and specialty physician care and emergency department visits among community-dwelling older adults with complex care needs

PLOS ONE

Dear Mr. Jones,

Thank you for submitting your manuscript to PLOS ONE. After careful consideration, we feel that it has merit but does not fully meet PLOS ONE’s publication criteria as it currently stands. Therefore, we invite you to submit a revised version of the manuscript that addresses the points raised during the review process.

This is a very interesting manuscript that addresses a relevant topic. Overall, the methods of the study are appropriate, the results are clearly presented and the discussion is well developed. However, there are some questions raised by the reviewers that should be responded.

Besides, I observe other minor points to be clarified:

I am not familiar with the organization of health care in Ontario. Readers from countries other than Canada will find that a brief description of the Ontario Home Care program and its criteria for qualification are helpful.Authors must explain how hospitalizations were managed in the study. There is not any mention about it in methods section. According to the manuscript and appendix 1, I assume that complete information about admissions is registered in the databases. So, I do not fully understand why there not were not included as a covariate nor considered a reason for censoring the outcome.The criteria to identify chronic conditions and medications should be described. Was any single record or prescription considered sufficient?The effect modification across multimorbidity and specialized care subpopulation is well presented in figure 1. However, there is a discrepancy with the text. According to the p-values in such figure, there are also statistically significant differences across categories of use of specialties in the high vs. med primary care.There is a typo in line 217. The HR 0,88 corresponds to the full model not primary care (table 3)

We would appreciate receiving your revised manuscript by Jun 07 2020 11:59PM. To enhance the reproducibility of your results, we recommend that if applicable you deposit your laboratory protocols in protocols.io, where a protocol can be assigned its own identifier (DOI) such that it can be cited independently in the future. For instructions see: http://journals.plos.org/plosone/s/submission-guidelines#loc-laboratory-protocols

We look forward to receiving your revised manuscript.

Kind regards,

Juan F. Orueta, MD, PhD

Academic Editor

PLOS ONE

Reviewers' comments:

Reviewer's Responses to Questions

**Comments to the Author**

1. Is the manuscript technically sound, and do the data support the conclusions?

Reviewer #1: Partly

Reviewer #2: Yes

2. Has the statistical analysis been performed appropriately and rigorously? 

Reviewer #1: No

Reviewer #2: Yes

3. Have the authors made all data underlying the findings in their manuscript fully available?

Reviewer #1: Yes

Reviewer #2: No

4. Is the manuscript presented in an intelligible fashion and written in standard English?

Reviewer #1: No

Reviewer #2: Yes

5. Review Comments to the Author

Reviewer #1: Continuity of care is a popular research topic. Literature has consistently shown that continuity of both primary care and specialty physician care improves health outcomes. Hence, what is the significance to exam it again on the same population which was emphasized on page 4 line 76. Please elaborate.

Please also explain why you choose the emergency care as the only outcome in your study? Is it more relevant to your research population? or is there any other reasons?

I think it would be easier to understand the advantage of your data profile by providing a flow chart that connects of all of your data sources and variables. (page 5 line 92)

Measures

I think it is reasonable to modify the original Bice-Boxerman score to make it more appropriate for measuring the long term continuity of care for patients with complex health specialty care needs. However, it is confusing for people who are not so familiar with the formula to comprehend the following two sentences (page 7 line 129- 131). It seems to contradict the earlier description (page 6 line 126- 127). I feel confuse and cannot make a logical connection with the following two paragraph Please provide more detail connection with that.

Page 7 line 151-152

Please explain what does “time” mean. Is it a continuous variable or a dichotomous variable? If there were a couple times of emergency visits, how do you define the dependent variable?

Page 12-13 line 215-218

The explanation of the line 215-218 is quite confusing and misleading.

In the full model, “high vs. medium” continuity of primary care was associated with an HR of 0.96 (95% CI 0.94-0.97) compared to a HR of 0.95 (95% CI 0.93-0.97) for specialty care.

Is “high vs. medium” a typo? I think it should be “medium vs. low” not “high vs. medium”.

I believe the HR of 0.96 was estimated from the comparison of medium vs. low continuity of care groups instead of high vs. medium groups.

Please clearly define the dependent variables of the model 1 and model 2. Does model 1 (primary care only model) only include patients who only see primary care physicians and never see specialty physicians? Or does it only include patients’ primary care visits and not their specialty care visits?

In addition, I wonder why coefficients on the “count of physician specialties seen” variable were included in the model 1 (primary care only model). Please explain it.

I cannot find fig.1.

Line 226-228.

The HR of high vs. low specialty care among 4+ specialties was 0.84 (0.81-0.86), while the HRs for patients who saw 3 specialties was 0.90 (0.87-0.93) and 2 specialties was 0.89 (0.86-0.92).

Please indicate all the data shown in this sentence to the table. I cannot find those data on table 3.

Page 14 Table 4

Please explain why there are empty coefficients in table 4. For example, there are no coefficients on the interaction variable of “Continuity of primary care * Count of chronic conditions” in model 1.

Please also explain why there are coefficients on the interaction term of Continuity of primary care * Count of chronic conditions in model 2 which suppose Specialty care only.

Reviewer #2: This is a study that examined the association between continuity of care and emergency visits. Overall, this is a well-conducted study, in a large, representative and well-characterized sample. The analyses have also been properly conducted. The findings will have high relevance to clinicians and policy-makers, as it will impact on how we structure our healthcare services, especially with regards to continuity of care.

I have several key concerns, that if addressed, I believe will greatly strengthen the manuscript and make the findings more convincing:

1) Methods: It will be good if the authors can show a timeline (in figure form) of which variable is captured at which time, paying particular attention to the time of baseline data & covariates, time of assessment for continuity, and time of outcome assessment.

2) Emergency Department Visits (page 7, line 150): It is unclear why the authors decided to only followup the patients for 6 months, considering that this is a rich dataset which could have provided much more information on the Emergency visits between 2016 to present. I will strongly suggest that the authors extend the followup period to longer than 6 months, and if possible, to the present time. This will improve the power in detecting meaningful interaction effects, which is also a key question of this manuscript.

3) Covariates (page 8, line 162): It is unclear how the authors decided on the specific chronic conditions. Also, why not based on established lists such as the Charlson Comorbidity Index?

4) Results (page 9, line 194): How was Mild cognitive impairment and activities of daily living (ADL) assessed? These information should be available in the Methods section. In addition, I also strongly suggest considering the interaction effects with cognitive impairment as well as with ADL. Apart from number of chronic diseases and number of specialists, cognitive impairment and ADL are also relevant interaction effects that reader will be interested in (especially considering how common they are in this sample, as well as in routine practices).

5) Discussion (page 16, line 277): it may be difficult to conclude on "no meaningful modification", until and unless we can satisfy that there has been sufficient power to do so (cf. point 2 above).

6) Figure 1: It is not clear what is meant by "High v. Med" and "High v. Low". I also wonder why the reference point became "High" in this figure, whereas the reference point was "Low" in the tables. I suggest only showing in the figure the significant interaction effects. For the non-significant ones, probably just mention them (together with the p values) in the text itself.

6. PLOS authors have the option to publish the peer review history of their article (what does this mean?). If published, this will include your full peer review and any attached files.

Reviewer #1: No

Reviewer #2: Yes: Dr. Tau Ming Liew

---

## [Author Response · Author response to Decision Letter 0]

5 May 2020

PONE-D-20-00490

Associations between continuity of primary and specialty physician care and emergency department visits among community-dwelling older adults with complex care needs

PLOS ONE

Dear Mr. Jones,

Thank you for submitting your manuscript to PLOS ONE. After careful consideration, we feel that it has merit but does not fully meet PLOS ONE’s publication criteria as it currently stands. Therefore, we invite you to submit a revised version of the manuscript that addresses the points raised during the review process.

This is a very interesting manuscript that addresses a relevant topic. Overall, the methods of the study are appropriate, the results are clearly presented and the discussion is well developed. However, there are some questions raised by the reviewers that should be responded.

Besides, I observe other minor points to be clarified:

I am not familiar with the organization of health care in Ontario. Readers from countries other than Canada will find that a brief description of the Ontario Home Care program and its criteria for qualification are helpful.

RESPONSE: 

We would like to thank the editor for their helpful comments and suggestions. We have added a description of Ontario’s home care system to the “Setting” section. (pg. 5). 

Authors must explain how hospitalizations were managed in the study. There is not any mention about it in methods section. According to the manuscript and appendix 1, I assume that complete information about admissions is registered in the databases. So, I do not fully understand why there not were not included as a covariate nor considered a reason for censoring the outcome.

RESPONSE: 

We do have data on hospital admissions and based on the comments of from the editor and reviewers we have decided to incorporate hospital admissions into our study in the following ways:

1. Add hospital admission as an outcome along with emergency department visit

 a. Simplify presentation of main results by showing only full models.

2. Add prior hospital admission as a covariate in models

We did not include hospital admission as a censoring variable because virtually all hospital admissions are preceded by an emergency department visit. Since we know that all patients are in the community at baseline, adding hospital admission as a censoring variable would not have an impact on the emergency department outcome since the ED visit would have happened first. 

Edits to the manuscript due to these analytic changes include updates to all sections of the manuscript, including title. There are been some minor changes in the discussion and conclusion section to better align with the new results.

The criteria to identify chronic conditions and medications should be described. Was any single record or prescription considered sufficient?

RESPONSE: 

The chronic conditions and use of medications were not extracted from administrative data but rather from the baseline RAI-HC clinical assessment, which has a single yes/no variable for each chronic condition and a count of the number of concurrent medications. We have updated the “Covariates” section (pg.9) to make this point clear. The reliability and validity of the RAI-HC is noted on pg. 6

The effect modification across multimorbidity and specialized care subpopulation is well presented in figure 1. However, there is a discrepancy with the text. According to the p-values in such figure, there are also statistically significant differences across categories of use of specialties in the high vs. med primary care.

RESPONSE: 

There are been significant changes to the effect modification section of the manuscript, but we have ensured that we report on every significant effect modification p-value.

There is a typo in line 217. The HR 0,88 corresponds to the full model not primary care (table 3)

RESPONSE: 

There are been significant changes to the results section of the manuscript, but we have checked to ensure that the HRs in the tables match to the manuscript.

5. Review Comments to the Author

Reviewer #1: Continuity of care is a popular research topic. Literature has consistently shown that continuity of both primary care and specialty physician care improves health outcomes. Hence, what is the significance to exam it again on the same population which was emphasized on page 4 line 76. Please elaborate.

RESPONSE: 

We would like to thank the reviewer for their helpful comments and suggestions.

Continuity of primary care has been well-explored in literature while continuity of specialty care has recently started to be explored. But there has been almost no research comparing the relative effectiveness of the two. The novelty of this study is that we concurrently look at both measures of continuity in a population that are significant users of both. Our results are important as they can guide how and which whom continuity of care should be promoted in older adults with complex care needs.We have added some additional text on pg.4 to underscore the novelty. 

Please also explain why you choose the emergency care as the only outcome in your study? Is it more relevant to your research population? or is there any other reasons?

RESPONSE:

Emergency department visits are an important outcome for our population given that home care patients have very high rates of ED use. We have updated the outcomes section of the analysis with this rationale. However, in response to comments from the editor and reviewers we have also added hospital admission as an additional outcome (pg. 8)

I think it would be easier to understand the advantage of your data profile by providing a flow chart that connects of all of your data sources and variables. (page 5 line 92)

RESPONSE:

We agree and have added a figure (Figure 1) that displays our timelines and when variables were measured.

Measures

I think it is reasonable to modify the original Bice-Boxerman score to make it more appropriate for measuring the long term continuity of care for patients with complex health specialty care needs. However, it is confusing for people who are not so familiar with the formula to comprehend the following two sentences (page 7 line 129- 131). It seems to contradict the earlier description (page 6 line 126- 127). I feel confuse and cannot make a logical connection with the following two paragraph Please provide more detail connection with that.

RESPONSE:

We agree that this point can be confusing for those who are not as familiar with the concepts and have added additional explanatory text to pgs 7-8.

Seeing more physician specialties naturally lowers continuity of care since physicians operating in those specialties are different providers. But if patients are indicated to see physicians from multiple specialties then they would likely benefit in terms of our outcomes. This is problematic as we generally hypothesize that better continuity results in better outcomes but in this case lower continuity would be associated with better outcomes. We derive our modified index to remove the decrease in continuity that comes from merely seeing multiple physician specialties, only considering the decrease in continuity that comes from seeing multiple physicians within the same specialty. 

Measures

Page 7 line 151-152

Please explain what does “time” mean. Is it a continuous variable or a dichotomous variable? If there were a couple times of emergency visits, how do you define the dependent variable?

RESPONSE:

We have replaced time with “days” and specified we measured until the first emergency department visit during follow-up. (pg. 9)

Page 12-13 line 215-218

The explanation of the line 215-218 is quite confusing and misleading.

In the full model, “high vs. medium” continuity of primary care was associated with an HR of 0.96 (95% CI 0.94-0.97) compared to a HR of 0.95 (95% CI 0.93-0.97) for specialty care.

Is “high vs. medium” a typo? I think it should be “medium vs. low” not “high vs. medium”.

I believe the HR of 0.96 was estimated from the comparison of medium vs. low continuity of care groups instead of high vs. medium groups.

RESPONSE:

The reviewer is correct there were mistakes in the manuscript with respect to “high vs. medium”. Low is the reference category for all comparisons, which take the form of “high vs. low” and “medium vs. low”. This has been rectified in this text. (pgs. 10-15)

Please clearly define the dependent variables of the model 1 and model 2. Does model 1 (primary care only model) only include patients who only see primary care physicians and never see specialty physicians? Or does it only include patients’ primary care visits and not their specialty care visits?

In addition, I wonder why coefficients on the “count of physician specialties seen” variable were included in the model 1 (primary care only model). Please explain it.

RESPONSE:

The three models were originally fit to demonstrate how the associations of continuity of primary and specialty care did not meaningfully change with the additional of the other into the model. All models were fit on the same population, which are patients with more primary and specialty care visits. However, after considering all comments from the editor and reviewers, we have decided to show only the full models as the results from these models are the that we use to draw conclusions. Hopefully this will prevent any future confusion.

The results section of the text has been updated accordingly. (pgs. 10-15)

I cannot find fig.1.

RESPONSE:

As per PLOS ONE guidelines figures are uploaded as separate files and are not embedded in the text.

Line 226-228.

The HR of high vs. low specialty care among 4+ specialties was 0.84 (0.81-0.86), while the HRs for patients who saw 3 specialties was 0.90 (0.87-0.93) and 2 specialties was 0.89 (0.86-0.92).

Please indicate all the data shown in this sentence to the table. I cannot find those data on table 3.

RESPONSE:

The effect modification section of the manuscript has been substantially reworked. We have checked to ensure that all stratum-specific HRs reported in the text match figures 2 and 3. To simply the effect modification analysis results we have removed the sensitivity analysis (the previous table 4) that looked at continuous interaction.

Page 14 Table 4

Please explain why there are empty coefficients in table 4. For example, there are no coefficients on the interaction variable of “Continuity of primary care * Count of chronic conditions” in model 1.

Please also explain why there are coefficients on the interaction term of Continuity of primary care * Count of chronic conditions in model 2 which suppose Specialty care only.

RESPONSE:

The previous table 4 showed the results of a sensitivity analysis in which three models were fit demonstrating progression from no interaction models to models with continuous interactions with count of chronic conditions and count of specialties. 

However, given how we have expanded our effect modification analysis to include the new hospital admission outcome and functional and cognitive impairment as effect modifiers, we have removed this sensitivity analysis as we feel that it was confusing and did not contribute meaningfully to the manuscript. Please see figures 2 and 3 for the full effect modification results.

Reviewer #2: This is a study that examined the association between continuity of care and emergency visits. Overall, this is a well-conducted study, in a large, representative and well-characterized sample. The analyses have also been properly conducted. The findings will have high relevance to clinicians and policy-makers, as it will impact on how we structure our healthcare services, especially with regards to continuity of care.

I have several key concerns, that if addressed, I believe will greatly strengthen the manuscript and make the findings more convincing:

1) Methods: It will be good if the authors can show a timeline (in figure form) of which variable is captured at which time, paying particular attention to the time of baseline data & covariates, time of assessment for continuity, and time of outcome assessment.

RESPONSE:

We would like to thank the reviewer for their helpful comments and suggestions.

We have added a figure 1 which describes the timeline of the study and when measurement of variables occurred.

2) Emergency Department Visits (page 7, line 150): It is unclear why the authors decided to only followup the patients for 6 months, considering that this is a rich dataset which could have provided much more information on the Emergency visits between 2016 to present. I will strongly suggest that the authors extend the followup period to longer than 6 months, and if possible, to the present time. This will improve the power in detecting meaningful interaction effects, which is also a key question of this manuscript.

RESPONSE:

Following up this particular group of patients for 6 months is standard practice given the instability of their health conditions. Previous research has demonstrated that there a high proportion of home care patients have a significant change in health status within 6 months (1), and therefore it is ideal to keep the follow-up time shorter so that our outcome window is closer to the baseline measurements. This does not result in low power as ~50% of home care patients have an emergency department visit within 6 months (49% in this manuscript) which is ideal for power. If we were to significantly lengthen the follow-up there is the potential to lose power as events would become more common than non-events.

Citation:

(1) Poss, J. Mind the gap? Looking at reassessment patterns among Ontario longstay home care clients. Proceedings from the 2009 Canadian interRAI Conference; 2009 June 22-24; Halifax (NS).

3) Covariates (page 8, line 162): It is unclear how the authors decided on the specific chronic conditions. Also, why not based on established lists such as the Charlson Comorbidity Index?

RESPONSE:

Congestive heart failure and chronic obstructive pulmonary disease are the two most important chronic diseases that influence the emergency department use of home care patients as has been shown in previous research. We have updated our covariate selection to state this. (pgs. 9-10).

In addition to CHF and COPD we have also looked at an overall count of chronic conditions. The Charlson Comorbidity index cannot be calculated from our baseline assessment, however our selected covariates include much of what is in the Charlson index: age, CHF, stroke, diabetes, dementia, COPD, cancer, and kidney disease.

4) Results (page 9, line 194): How was Mild cognitive impairment and activities of daily living (ADL) assessed? These information should be available in the Methods section. In addition, I also strongly suggest considering the interaction effects with cognitive impairment as well as with ADL. Apart from number of chronic diseases and number of specialists, cognitive impairment and ADL are also relevant interaction effects that reader will be interested in (especially considering how common they are in this sample, as well as in routine practices).

RESPONSE:

We measured functional impairment with the Activities of Daily Living Hierarchy and cognitive impairment with the Cognitive Performance Scale. The text has been updated with this information. (pg. 10)

We have updated our effect modification analysis to include an examination of modification by functional and cognitive impairment. We have updated the methods, results, and discussion of our manuscript with these changes.

5) Discussion (page 16, line 277): it may be difficult to conclude on "no meaningful modification", until and unless we can satisfy that there has been sufficient power to do so (cf. point 2 above).

RESPONSE:

We would argue that we have sufficient power to discover clinically meaningful interactions based on our large sample size (178,000) , high event rate (~50%), and demonstrated detection of interactions where hazard ratios differ by only ~0.05. For example, in figure 2, high vs. low continuity of specialty care has HRs of 0.94, 0.96, and 0.90 across the 2, 3, and 4+ specialties, with an overall p-value of 0.026. Also, in figure 2, high vs. low continuity of primary care has HRs of 0.89, 0.93, and 0.93 across CPS categories of 0-1, 2-3, and 4-6, with an overall p-value of 0.030

6) Figure 1: It is not clear what is meant by "High v. Med" and "High v. Low". I also wonder why the reference point became "High" in this figure, whereas the reference point was "Low" in the tables. I suggest only showing in the figure the significant interaction effects. For the non-significant ones, probably just mention them (together with the p values) in the text itself.

RESPONSE:

We would like to thank the reviewer for pointing out these mistakes. They have been rectified in the new tables and figures. Low is now the consistent reference category.

---

## [Decision Letter · Decision Letter 1]

21 May 2020

Associations between continuity of primary and specialty physician care and use of hospital-based care among community-dwelling older adults with complex care needs

PONE-D-20-00490R1

Dear Dr. Jones,

We are pleased to inform you that your manuscript has been judged scientifically suitable for publication and will be formally accepted for publication once it complies with all outstanding technical requirements.

With kind regards,

Juan F. Orueta, MD, PhD

Academic Editor

PLOS ONE

Additional Editor Comments :

My personal opinion is that the authors have greatly improved the manuscript and produced a notable paper. I would like to congratulate them for their excellent work

Otherwise, I have spotted two typos. Please correct them:

• In Abstract: Line#41 the HR for hospital admission should be “HR=0.94 (0.92-0.96)” (instead of “HR=0.94 (0.82-0.96)”)

• In Figure 2: The value for Primary Care, All patients (the first row) should be “0.90 (0.89-0.92)” (instead of “0.96 (0.94-0.97)”)

Reviewers' comments:

Reviewer's Responses to Questions

**Comments to the Author**

1. If the authors have adequately addressed your comments raised in a previous round of review and you feel that this manuscript is now acceptable for publication, you may indicate that here to bypass the “Comments to the Author” section, enter your conflict of interest statement in the “Confidential to Editor” section, and submit your "Accept" recommendation.

Reviewer #1: All comments have been addressed

2. Is the manuscript technically sound, and do the data support the conclusions?

Reviewer #1: Yes

3. Has the statistical analysis been performed appropriately and rigorously? 

Reviewer #1: Yes

4. Have the authors made all data underlying the findings in their manuscript fully available?

Reviewer #1: Yes

5. Is the manuscript presented in an intelligible fashion and written in standard English?

Reviewer #1: Yes

6. Review Comments to the Author

Reviewer #1: The authors have responded appropriately to my review comments and made the revision more than I expected. I have no further questions.

7. PLOS authors have the option to publish the peer review history of their article (what does this mean?). If published, this will include your full peer review and any attached files.

Reviewer #1: Yes: Hui-Chu Lang, Ph.D.

---

## [Editor Report · Acceptance letter]

29 May 2020

PONE-D-20-00490R1 

Associations between continuity of primary and specialty physician care and use of hospital-based care among community-dwelling older adults with complex care needs 

Dear Dr. Jones:

I am pleased to inform you that your manuscript has been deemed suitable for publication in PLOS ONE. Congratulations! Your manuscript is now with our production department. 

With kind regards,

on behalf of

Dr. Juan F. Orueta 

Academic Editor

PLOS ONE